# Structure of PINK1 and mechanisms of Parkinson's disease-associated mutations

Atul Kumar[1,2†], Jevgenia Tamjar[2†], Andrew D Waddell[2†], Helen I Woodroof[2], Olawale G Raimi[1], Andrew M Shaw[2], Mark Peggie[3], Miratul MK Muqit[2,4*], Daan MF van Aalten[1*]

[1]Division of Gene Regulation and Expression, University of Dundee, Dundee, United Kingdom; [2]MRC Protein Phosphorylation and Ubiquitylation Unit, University of Dundee, Dundee, United Kingdom; [3]Division of Signal Transduction Therapy, School of Life Sciences, University of Dundee, Dundee, United Kingdom; [4]School of Medicine, University of Dundee, Dundee, United Kingdom

**Abstract** Mutations in the human kinase PINK1 (hPINK1) are associated with autosomal recessive early-onset Parkinson's disease (PD). hPINK1 activates Parkin E3 ligase activity, involving phosphorylation of ubiquitin and the Parkin ubiquitin-like (Ubl) domain *via* as yet poorly understood mechanisms. hPINK1 is unusual amongst kinases due to the presence of three loop insertions of unknown function. We report the structure of *Tribolium castaneum* PINK1 (*Tc*PINK1), revealing several unique extensions to the canonical protein kinase fold. The third insertion, together with autophosphorylation at residue Ser205, contributes to formation of a bowl-shaped binding site for ubiquitin. We also define a novel structural element within the second insertion that is held together by a distal loop that is critical for *Tc*PINK1 activity. The structure of *Tc*PINK1 explains how PD-linked mutations that lie within the kinase domain result in hPINK1 loss-of-function and provides a platform for the exploration of small molecule modulators of hPINK1.
DOI: https://doi.org/10.7554/eLife.29985.001

*For correspondence:
m.muqit@dundee.ac.uk (MMKM);
dmfvanaalten@dundee.ac.uk
(DMFA)

†These authors contributed
equally to this work

Competing interests: The
authors declare that no
competing interests exist.

Reviewing editor: Tony Hunter,
Salk Institute for Biological
Studies, United States

## Introduction

Autosomal recessive inherited loss-of-function mutations in human PTEN-induced kinase 1 (hPINK1), represent the second most frequent cause of early-onset Parkinson's disease (PD) (*Valente et al., 2004*). hPINK1 has been proposed to act as a master regulator of mitochondrial quality control, promoting the elimination of damaged mitochondria via autophagy known as mitophagy (*McWilliams and Muqit, 2017*). In response to mitochondrial depolarisation, hPINK1 is activated and phosphorylates ubiquitin at Serine65 (Ser65). Ser65-phosphorylated ubiquitin then binds to the ubiquitin E3 ligase Parkin with high affinity, serving to prime Parkin for phosphorylation by hPINK1 at an equivalent Ser65 residue that lies within its N-terminal ubiquitin-like domain (Ubl) (*Kondapalli et al., 2012*; *Kazlauskaite et al., 2014*; *Kane et al., 2014*; *Koyano et al., 2014*). This chain of phosphorylation events stimulates maximal activation of Parkin E3 ligase activity, resulting in ubiquitylation of multiple substrates at the outer mitochondrial membrane and represents a critical upstream step in the induction of mitophagy (*McWilliams and Muqit, 2017*). The molecular mechanisms of hPINK1 activation and substrate recognition are poorly understood. hPINK1 is distinct from other protein kinases due to the presence of three unique insertions (Ins1, Ins2 and Ins3) within the kinase domain and a C-terminal extension (CTE) of unknown function that bears no homology to any known protein domain (*Figure 1A*). Nearly, 30 missense and nonsense hPINK1 mutations have been reported in patients worldwide (*Deas et al., 2009*), located predominantly within the kinase domain or predicted to truncate the CTE. However, the molecular mechanism by which they disrupt hPINK1 catalysis and/or recognition of its physiological substrates ubiquitin and Parkin is unknown. We have

**eLife digest** Estimates suggest that more than 10 million people worldwide are living with Parkinson's disease. This condition, which is incurable, is characterised by the death of brain cells leading to tremors and loss of motor control. So far scientists have been able to link mutations in nearly 20 different genes to Parkinson's disease. This includes the gene that codes for an enzyme called PINK1. Mutations that affect PINK1 are often seen in people with early-onset Parkinson's disease and cases that are inherited through families.

Changes to PINK1 affects the ability of cells to keep producing energy. Without enough energy to power biochemical processes, affected cells will eventually die. Understanding why PINK1 has this effect and how to prevent it could be helpful in treating Parkinson's disease. One way to understand the role of PINK1 is to study the three-dimensional structure of the protein to examine how it interacts with other molecules. The PINK1 protein from the flour beetle *Tribolium castaneum* is easier to produce than human PINK1; this makes it easier to study its structure in the laboratory.

Using flour beetle PINK1 and a technique called X-ray crystallography, Kumar, Tamjar, Waddell et al. made a detailed three-dimensional model of the structure of the PINK1 protein. This revealed new details of the structure including a region at one end called the "C-terminal extension", which is involved in PINK1's enzyme activities. The structure also highlighted the purposes of two loop regions, one that controls PINK1 activity and one that affects its interactions with other proteins. Further examination of some of these features showed that they are also part of human PINK1 proteins.

Using this new protein structure, Kumar et al. went on to examine 20 mutations that are found in patients with Parkinson's disease and have been able to show the effect that these changes have on PINK1 at a molecular level. It may now be possible to begin designing drugs to prevent or reverse these changes, which could lead to new treatments for Parkinson's disease.

DOI: https://doi.org/10.7554/eLife.29985.002

previously identified a hPINK1 orthologue in *Tribolium castaneum* (*Tc*PINK1), which in contrast to human hPINK1, almost entirely lacks the first kinase domain insertion (*Woodroof et al., 2011*) (*Figure 1A*). *Tc*PINK1 exhibits constitutive catalytic activity towards ubiquitin, Parkin and generic substrates in vitro (*Woodroof et al., 2011*; *Kazlauskaite et al., 2015*). Furthermore, the motor defects of *Drosophila* PINK1 null flies can be efficiently rescued in vivo by crossing lines that over-express *Tc*PINK1 (*Woodroof et al., 2011*). Herein, we report the crystal structure of *Tc*PINK1, revealing structural insights into the CTE and kinase domain loop insertions. In particular, we show that the third insertion contributes to formation of a bowl-shaped binding site for ubiquitin that is critical for either *Tc*PINK1 or hPINK1 mediated ubiquitin and Parkin Ubl phosphorylation. Furthermore, we have elaborated an important regulatory role for *Tc*PINK1 autophosphorylation of Serine205 (Ser205), which together with the third insertion, aids in physiological substrate recognition. Overall, these findings provide molecular insights into the mechanisms of hPINK1 kinase activity, ubiquitin substrate recognition, and define the molecular basis of Parkinson's disease-causing mutations.

## Results and discussion

### The TcPINK1 kinase fold is decorated with unique structural features

The crystal structure of a catalytically active fragment of *Tc*PINK1 was solved and refined to 2.78 Å using *E. coli* expression of a cDNA construct encompassing the kinase domain and CTE (Ser150 - Asp570), and possessing mutations to reduce surface entropy (Glu527Ala and Lys528Ala), a 10 amino acid deletion in a large loop (Δ261–270), and a phospho-mimetic mutation (Ser205Glu) (*Figure 1A*). The crystal structure of *Tc*PINK1 reveals core secondary structure elements of a typical protein kinase domain (*Hanks and Hunter, 1995*) (*Figure 1B,C*). As described previously (*Woodroof et al., 2011*), the N-lobe of the *Tc*PINK1 kinase domain contains three insertions: Lys182–Pro192 (Ins1), Glu221–Ala253 (Ins2), and Leu260–Met288 (Ins3) and in contrast to hPINK1, Ins1 exists as a loop remnant in *Tc*PINK1 (*Figure 1A,B*). Ins2 forms a small domain containing a βi-

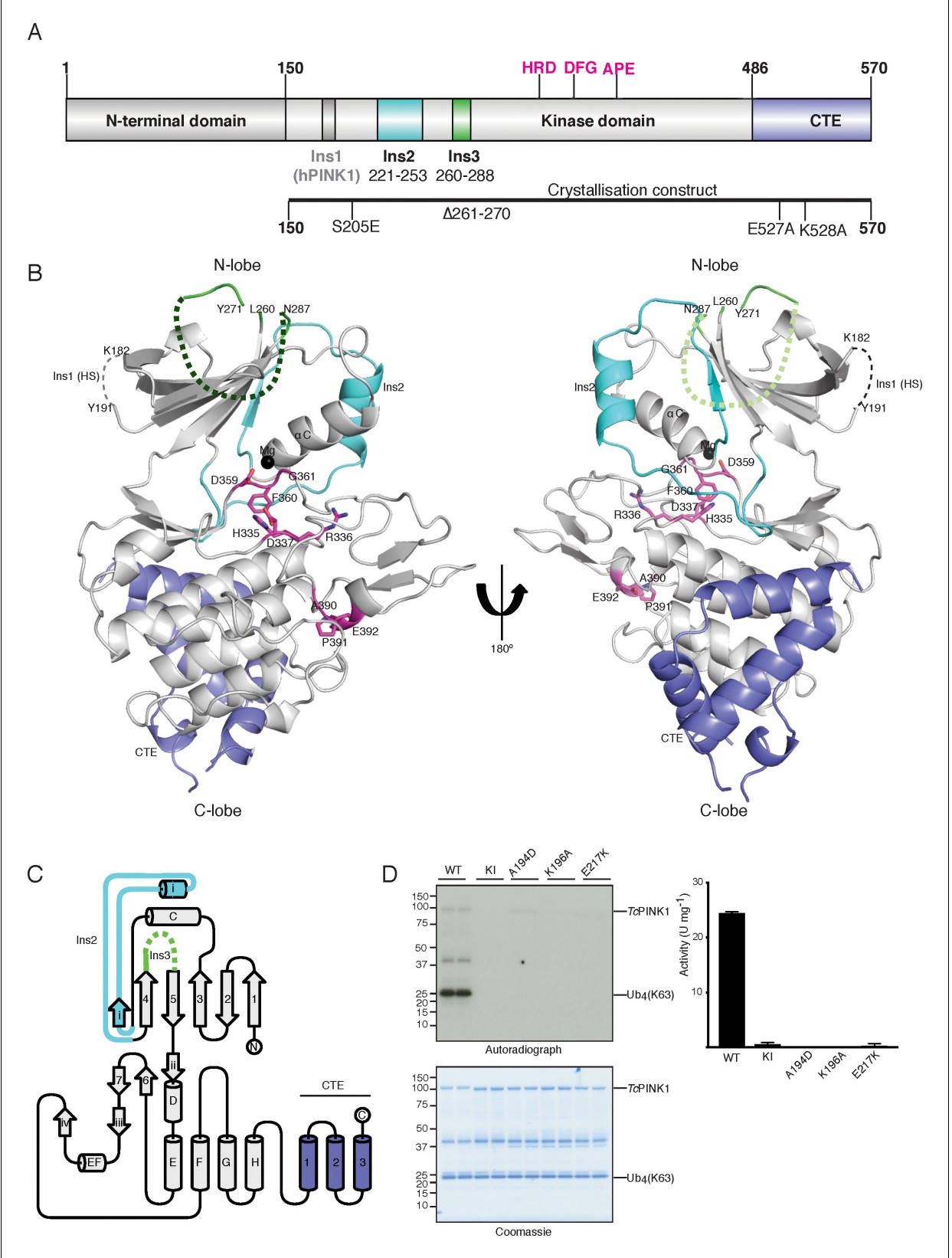

**Figure 1.** Overall structure of *Tc*PINK1. (**A**) Domain architecture of *Tc*PINK1 showing kinase domain (150-486); location of Ins2, Ins3, catalytic motif (HRD) and activation loop motifs (DFG; APE) and CTE (487-570). The location of hPINK1 Ins1, which is largely absent in *Tc*PINK1, is indicated. CTE,

*Figure 1 continued on next page*

Figure 1 continued

C-terminal extension; Ins, insertion. Created with IBS (*Liu et al., 2015*). The boundaries of the crystallisation construct with the mutations introduced is also shown, lower panel. (**B**) Overall structure of *Tc*PINK1 showing the canonical N-lobe and C-lobe (grey). Catalytic and activation loop motifs are shown in magenta sticks and $Mg^{2+}$ is shown as a black sphere. Ins2, Ins3 and CTE of *Tc*PINK1 are coloured as in *Figure 1A*; disordered regions of PINK1 are shown in dashed line; Ins1 of hPINK1 is also shown as part of disordered loop in *Tc*PINK1. (**C**) Topology diagram of the *Tc*PINK1 structure. Core kinase domain secondary structure elements are shown in grey and indicated with numbers (β-sheet) and letters (α-helices) according to the nomenclature in (*Hanks and Hunter, 1995*). Non-canonical secondary structure elements (CTE, C-terminal extension; Ins, insertion) are indicated with Roman numerals for Ins2 (cyan), Ins3 (green) and numbering for CTE (blue). Created with TopDraw (*Bond, 2003*). (**D**) Ubiquitin phosphorylation assay with full-length *Tc*PINK1 (WT), kinase-inactive D359A mutant (KI), and ATP-binding pocket mutations A194D, K196A and E217K. 2 µg of indicated enzyme was incubated with 2 µM ubiquitin (K63-linked tetraubiquitin) and [γ-$^{32}$P] ATP for 10 min. Gels were analysed by Coomassie staining (lower panel) and [γ-$^{32}$P] incorporation visualised by autoradiography (upper panel) followed by Cerenkov counting of substrate bands for quantification (right panel).

DOI: https://doi.org/10.7554/eLife.29985.003

The following figure supplement is available for figure 1:

**Figure supplement 1.** Mutations in the *Tc*PINK1 ATP binding pocket result in loss of kinase activity.

DOI: https://doi.org/10.7554/eLife.29985.004

strand (222-226) and an αi-helix (231-242) followed by a loop (243-253). Ins3 is disordered and is not included in the final model (*Figure 1B,C*). Finally, the majority of the *Tc*PINK1 CTE (Ala487–Leu556) is α-helical in nature (*Figure 1B,C*), and rather than forming a separate domain, is tightly packed on the E, G and H α-helices of the C-lobe of the kinase domain (3747 Å$^2$ buried surface area), supporting previous data on the importance of this region for catalysis (*Woodroof et al., 2011*). A DALI structure similarity search revealed that Ins2 and the CTE structural elements are unique to *Tc*PINK1 and not found in the almost thousand kinase structures solved to date. Although *Tc*PINK1 was crystallised without ATP, the kinase domain appears to adopt an active, closed conformation, based on the positioning of the αC helix, activation segment conformation, and the 'in' position of the DFG motif (*Hari et al., 2013*; *Huse and Kuriyan, 2002*; *Nolen et al., 2004*) (*Figure 1B*). A model of the ATP was included by superposition with the conformationally similar cAMP dependent protein kinase-ATP complex (RMSD 3.2 Å on 252 equivalent Cα atoms). This suggested that residues Ala194, Lys196, Leu344, Tyr297 and Glu217 are critical for ATP binding (*Figure 1—figure supplement 1A*). In agreement with this, biochemical analysis revealed that the Ala194Asp, Lys196Ala and Glu217Lys *Tc*PINK1 mutants prevented phosphorylation of its physiological substrates, ubiquitin and the Parkin Ubl domain (*Figure 1D*, *Figure 1—figure supplement 1B*). Furthermore, inspection of the structure indicates that the conformation of residues forming two conserved hydrophobic networks (the 'R-spine' comprising residues Ile253, Val219, Phe360, His335 and Asp410, and the 'C-spine' residues Val176, Leu301, Leu343, Leu344, Leu345, ILe421 and Ile417) is compatible with an active conformation (*Kornev et al., 2006*, *2008*).

## Ins3 forms the ubiquitin binding pocket

hPINK1-dependent phosphorylation of Ser65 on ubiquitin and the Ubl domain of Parkin is critical for maximal Parkin activation (*Wauer et al., 2015*; *Kumar et al., 2017*). However, the regions of hPINK1 that recognize and bind its substrates remain unknown. Via DALI structure similarity search, we identified the LIM kinase-cofilin complex (*Hamill et al., 2016*) as being conformationally similar to *Tc*PINK1 (RMSD 2.8 Å on 298 equivalenced Cα atoms). Using the phosphopeptide in this complex as an anchor (*Figure 2A*), we superimposed Ser65 of ubiquitin onto the *Tc*PINK1 active site, resulting in a model of the *Tc*PINK1-ubiquitin complex, free of steric clashes (*Figure 2B*). Using this model, we explored the key structural elements in *Tc*PINK1 that may contribute to recognition of ubiquitin. Although Ins3 in the *Tc*PINK1 kinase domain is mostly disordered and could not be built as part of the structure; its predicted location places it proximal to the ubiquitin in the model of the complex and therefore we hypothesized that it is likely to contribute a critical role towards substrate recognition and binding. To test this, we deleted 10 residues (residues 261–270, equivalent to the deletion used in the crystallisation construct) of Ins3 and measured *Tc*PINK1 catalytic activity using multiple physiological readouts, including *Tc*PINK1 catalysed phosphorylation of ubiquitin and the Parkin Ubl domain, and transphosphorylation of a kinase-inactive (D359A) *Tc*PINK1 fragment (residues 125–570; His-SUMO cleaved) (*Figure 2C,D*, *Figure 2—figure supplement 1A*). Strikingly,

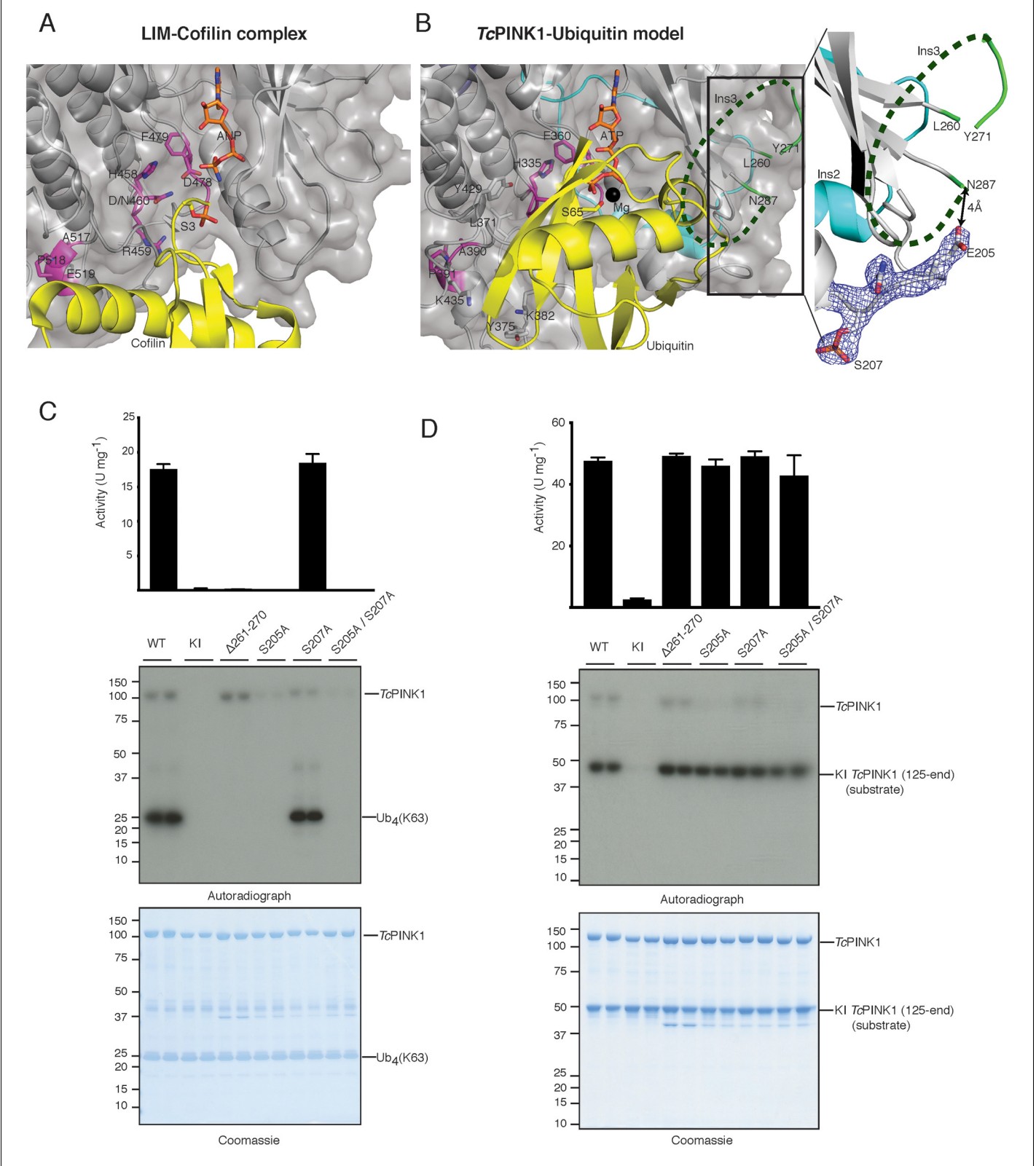

**Figure 2.** Identification of the PINK1 substrate binding bowl. (**A**) Crystal structure of the LIM kinase substrate complex (pdb id: 5HVK, 18) showing LIM kinase (coloured as *Tc*PINK1 in *Figure 1B*) and bound substrate cofilin (yellow) with Ser3 (yellow stick) poised in acceptor position. (**B**) Ubiquitin (yellow) was modelled in *Tc*PINK1 (coloured as in *Figure 1B*) by superposing Ser3 of cofilin with Ser65 (shown as sticks) of ubiquitin after superposing the

*Figure 2 continued on next page*

*Figure 2 continued*

*Tc*PINK1 and LIM kinase domains; disordered Ins3 is marked by dashed green line; 2F$_o$-F$_c$ map, contoured at 1.5σ, corresponding to phosphorylated Ser207 region is shown highlighting the Glu205 position in respect to Asn287 (tail of disordered Ins3), inset. (C) Ubiquitin phosphorylation assay with full-length *Tc*PINK1 (WT), kinase-inactive D359A mutant (KI), insertion three deletion mutant (Δ261–270), S205A and/or S207A mutants in the autophosphorylation sites. 2 μM ubiquitin (K63-linked tetraubiquitin) was used as substrate; assay conditions were similar to **Figure 1D**. Gels were analysed by Coomassie staining (lower panel) and [γ-$^{32}$P] incorporation visualised by autoradiography (middle panel) followed by Cerenkov counting of substrate bands for quantification (upper panel). (D) *Tc*PINK1 transphosphorylation assay with full-length *Tc*PINK1 (WT) or various mutants as in **Figure 2C**. 2 μM of kinase inactive *Tc*PINK1 (125-end) was used as substrate; assay conditions were similar to **Figure 1D**. Gels were analysed by Coomassie staining (lower panel) and [γ-$^{32}$P] incorporation visualised by autoradiography (middle panel) followed by Cerenkov counting of substrate bands for quantification (upper panel).

DOI: https://doi.org/10.7554/eLife.29985.005

The following figure supplement is available for figure 2:

**Figure supplement 1.** *Tc*PINK1 Mutations in Ins3, autophosphorylation site and canonical substrate binding region result in loss of activity.

DOI: https://doi.org/10.7554/eLife.29985.006

deletion of Ins3 resulted in complete loss of ubiquitin or Parkin Ubl phosphorylation, while preserving transphosphorylation activity towards kinase-inactive *Tc*PINK1 (**Figure 2C,D**, **Figure 2—figure supplement 1A**). We also found that mutation of key residues Leu371, Tyr375, Lys382 and Lys435 within the canonical substrate binding pocket of kinases resulted in loss of activity against ubiquitin (**Figure 2B**, **Figure 2—figure supplement 1B**).

We have previously identified multiple autophosphorylation sites for *Tc*PINK1 by mass spectrometry, including Ser205 (equivalent to Ser228 in hPINK1) and Ser207 (equivalent to Ser230 in hPINK1) (**Woodroof et al., 2011**). Ser207 phosphorylation was also observed in the crystal structure (**Figure 2B**). Inspection of the structure revealed that Glu205 (the phosphomimetic of Ser205) lies only 4 Å away from the tail (Asn287) of the disordered Ins3 (**Figure 2B**). We therefore hypothesized that autophosphorylation of Ser205 may influence and/or contribute to ubiquitin recognition and binding mediated by Ins3. Therefore, to test the effect of Ser205 phosphorylation in substrate recognition, we performed parallel assays with ubiquitin, Parkin Ubl domain and transphosphorylation of kinase-inactive *Tc*PINK1 as performed for the Ins3 deletion. Critically Ser205Ala, but not Ser207Ala, prevented ubiquitin and Parkin Ubl phosphorylation while preserving transphosphorylation activity towards kinase-inactive *Tc*PINK1 (**Figure 2C,D**, **Figure 2—figure supplement 1A**). Overall, these findings indicate that Ins3, unique to the PINK1 kinases, together with autophosphorylation at Ser205, is critical for ubiquitin and Parkin Ubl phosphorylation and may form part of the ubiquitin/Ubl fold recognition pocket.

## Role of novel regulatory features of TcPINK1 catalysis

Multiple lines of evidence suggest that hPINK1 is autoinhibited under basal conditions, including lack of significant activity of hPINK1 in vitro (**Woodroof et al., 2011**) and the detection of robust activity of hPINK1 in cells only under conditions of mitochondrial depolarization triggering hPINK1 activation *via* an unknown mechanism (**Kondapalli et al., 2012**). In contrast to hPINK1, insect orthologues isolated from *Drosophila melanogaster*, *Pediculus humanus corporis* and *Tribolium castaneum* exhibit constitutive catalytic activity in vitro (**Woodroof et al., 2011**; **Wauer et al., 2015**). Since the *Tc*PINK1 kinase domain adopts an active conformation in our crystal structure, we sought to exploit this knowledge to probe possible mechanisms of hPINK1 activation. Activation loop phosphorylation is the most common mechanism of protein kinase activation (**Hanks and Hunter, 1995**), with phosphorylation stabilising the active conformation via electrostatic interaction between the phospho-serine/threonine residue and a basic pocket consisting of the arginine in the HRD motif (**Nolen et al., 2004**). Previous mass spectrometry analysis of *Tc*PINK1 did not identify activation loop phosphorylation, and furthermore, alanine mutation of a candidate T-loop phosphorylation site residue, Ser377, did not impair *Tc*PINK1 catalytic activity (**Woodroof et al., 2011**). Consistent with this, we do not observe phosphorylation of Ser377 in the *Tc*PINK1 structure and furthermore, Ser377 is located too distant (12.2 Å) from Arg336 of the HRD motif to form an electrostatic interaction (**Figure 3A**). Instead the *Tc*PINK1 structure reveals that a highly conserved aspartate (Asp381) within the activation loop, an arginine residue (Arg336) of the HRD motif and an arginine residue (Arg216) in the αC-helix, form an equivalent interaction (**Figure 3A**), which is further stabilized via

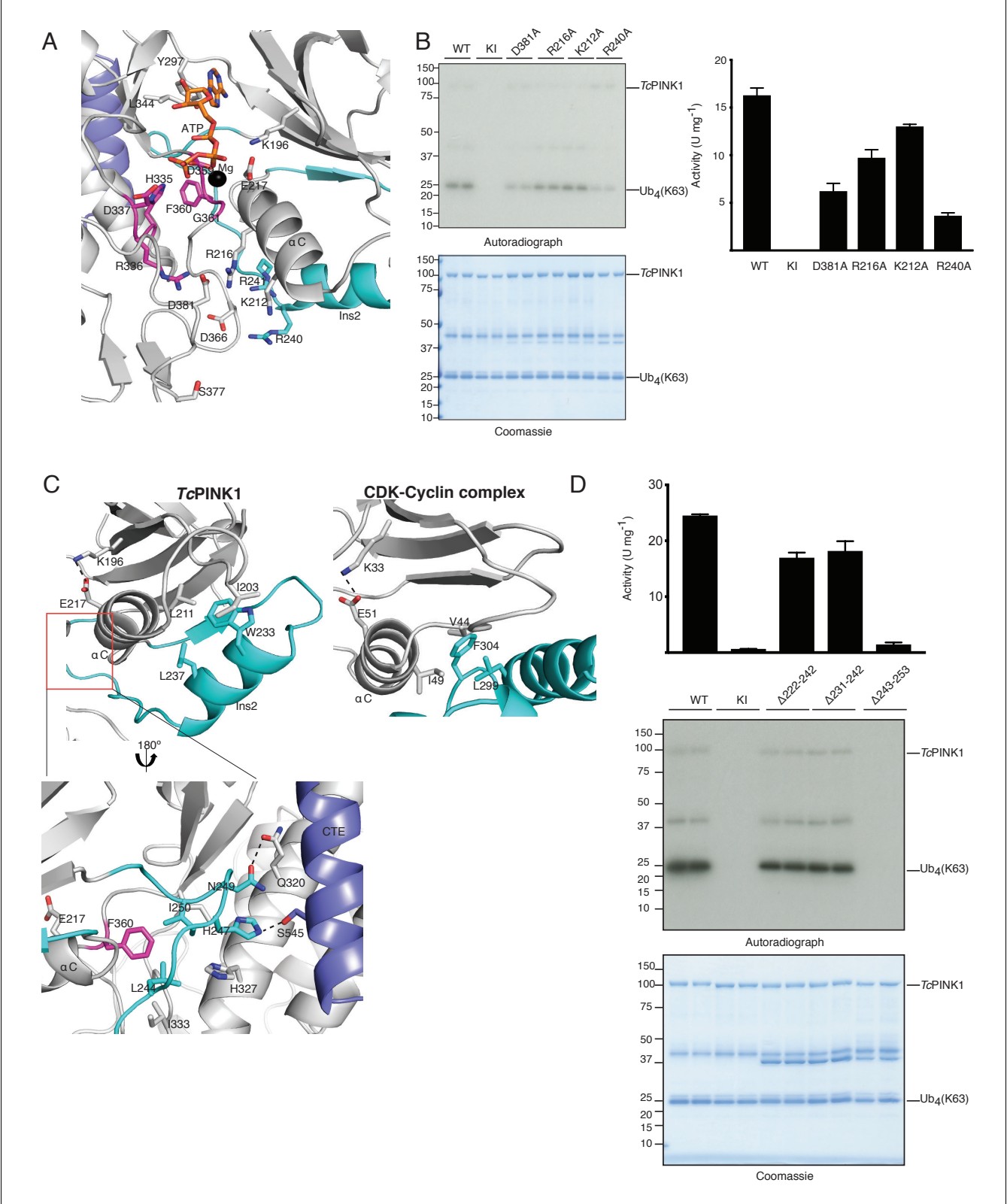

**Figure 3.** Ins2 of *Tc*PINK1 adopts a unique conformation. (**A**) Close up view of active site of *Tc*PINK1 (coloured as in *Figure 1B*) showing key ionic interactions involving Asp381 of the activation loop with Arg336 of the HRD motif and Arg216 of the αC helix; and Asp366 with Lys212 of the αC helix and Arg240 of Ins2. (**B**) Ubiquitin phosphorylation assay with full-length *Tc*PINK1 (WT), kinase-inactive D359A mutant (KI), D381A, R216A, K212A and R240A mutants. 2 μM of Ubiquitin (K63-linked tetraubiquitin) was used as substrate; conditions were similar to *Figure 1D*. Gels were analysed by

*Figure 3 continued on next page*

*Figure 3 continued*

Coomassie staining (lower panel) and [γ-$^{32}$P] incorporation visualised by autoradiography (upper panel) followed by Cerenkov counting of substrate bands for quantification (right panel). (**C**) Structural comparison of *Tc*PINK1 and the CDK-Cyclin complex (pdb id: 5IF1) showing intramolecular interactions between Ins2 (cyan) and αC helix (grey) of *Tc*PINK1 that resemble the intermolecular interactions between CDK (grey) and Cyclin (cyan). The salt bridge between Glu and Lys (shown in sticks) for kinase activation is represented by a dashed line. Interactions between the C-lobe and the distal Ins2 loop are highlighted (Inset). (**D**) Ubiquitin phosphorylation assay with full-length *Tc*PINK1 (WT), kinase-inactive D359A mutant (KI) and *Tc*PINK1 Ins2 mutants with βi-strand and αi-helix deletion (Δ222–242), αi-helix deletion (Δ231–242), or distal loop deletion (Δ243–253). 2 μM of Ubiquitin (K63-linked tetraubiquitin) was used as substrate, conditions were similar to *Figure 1D*. Gels were analysed by Coomassie staining (lower panel) and [γ-$^{32}$P] incorporation visualised by autoradiography (middle panel) followed by Cerenkov counting of substrate bands for quantification (upper panel).

DOI: https://doi.org/10.7554/eLife.29985.007

The following figure supplement is available for figure 3:

**Figure supplement 1.** High conservation and functionally important role of C-terminal distal loop of Ins2 in *Tc*PINK1 activity.

DOI: https://doi.org/10.7554/eLife.29985.008

interaction with basic residues (Lys212 on the αC-helix; Arg240 and Arg241 in Ins2) and another conserved aspartate (Asp366). To explore which residues were important for *Tc*PINK1 catalysed phosphorylation of its substrates, we investigated point mutants of residues lying within this region. The Asp366Ala *Tc*PINK1 mutant was unstable under our expression conditions and was not assessed. Importantly, our data revealed that mutants in this region, particularly the Asp381Ala and Arg240Ala mutants, were associated with a significant reduction in *Tc*PINK1 catalytic kinase activity towards ubiquitin (*Figure 3B*). Although Asp381 and Asp366 are conserved in hPINK1, Lys212 and Arg216 form a basic patch on the αC-helix that is conserved among the insect orthologues but not in hPINK1, where the equivalent residues are Asn235 and Gln239. Furthermore, Arg240 and Arg241 of *Tc*PINK1 are replaced by Gly264 and Pro265 in hPINK1, respectively (*Figure 3—figure supplement 1A*). We hypothesized that these differences in basic residues between *Tc*PINK1 and hPINK1 could explain the lack of recombinant hPINK1 activity. To test this, we generated a mutant of hPINK1 incorporating Asn235Lys/Gln239Arg/Gly264Arg/Pro265Arg substitutions, however, this did not lead to any significant enhancement in hPINK1 activity (data not shown).

Inspection of the *Tc*PINK1 structure suggests another potential regulatory mechanism of activation through interaction of Ins2 with the αC-helix (1120 Å$^2$ buried surface area) (*Figure 3C*). This interaction is reminiscent of the activation mechanism of cyclin dependent kinases (CDKs), whereby intermolecular stabilization of the CDK αC-helix occurs due to interactions with the cyclin α5-helix (*Figure 3C*) (*Jeffrey et al., 1995*). The *Tc*PINK1 αi-helix (residues 231–242) of Ins2 (residues 221–253) occupies a similar position to cyclin, inducing a displacement of the αC-helix and allowing formation of the key salt bridge between the conserved Glu217 on the αC-helix and the conserved Lys196 on the N-lobe, maintaining the kinase in an active conformation (*Figure 3C*). Akin to the intermolecular CDK-cyclin interaction, the *Tc*PINK1 αC-helix-αi-helix intramolecular interface also consists of hydrophobic interactions (*Figure 3C*). Whilst the overall conservation of Ins2 (residues 221–253) is low, the C-terminal/distal region (243-253) of Ins2 remains highly conserved (*Figure 3—figure supplement 1A*), forming a loop (Ins2 loop) which tethers the proximal region of Ins2 (βi-strand and αi-helix) with the rest of the C-lobe mediated by interactions between His247 and Asn249 of Ins2 and Ser545 of CTE and Gln320 of the C-lobe, respectively. Furthermore, a hydrophobic core is formed between Leu244 and Ile250 of Ins2 with Phe360 of the 'DFG motif' and Ile357 (*Figure 3C*). Interestingly, deletion of the αi-helix (residues 231–242) alone or in combination with the βi-strand (residues 222–242) region of Ins2 does not affect the catalytic activity of *Tc*PINK1. However, deletion of the Ins2 loop (residues 243–253) abolishes the kinase activity of *Tc*PINK1 (*Figure 3D*, *Figure 3—figure supplement 1B*). Therefore, our analysis suggests a critical role for the Ins2 loop (243-253) as an anchor, to maintaining the kinase domain in an active state, although the function of the proximal region of Ins2 (222-242) remains to be fully elucidated.

## Translating structural insights to hPINK1

Sequence alignment of hPINK1 and *Tc*PINK1 reveals an overall high degree of conservation with the exception of the relative absence of Ins1 in *Tc*PINK1 (*Figure 4—figure supplement 1*). Therefore, we next explored the structural and functional conservation between insect and human PINK1 in cells expressing hPINK1 whose kinase activity can be stimulated by carbonyl cyanide

m-cholorophenylhydrazone (CCCP) that induces mitochondrial depolarization (*Kazlauskaite et al., 2015*). We transiently expressed wild type hPINK1 or mutants of key ATP-binding residues predicted from the *Tc*PINK1 structure (*Figure 1D*, *Figure 1—figure supplement 1A*) into HeLa hPINK1 knock-out cells generated by TALEN technology (*Narendra et al., 2013*). Cells were treated with 10 µM CCCP or DMSO for 3 hr and extracts immunoblotted with phospho-specific antibodies raised against hPINK1 substrates Ser65 Parkin and Ser65 ubiquitin. Consistent with our analysis of *Tc*PINK1 (*Figure 1D*), we found that Parkin and ubiquitin phosphorylation was abolished in cells expressing mutant A217D, K219A, E240K and L369P forms of hPINK1 upon CCCP stimulation (*Figure 4A*). This indicates that the ATP-binding pocket is well conserved between *Tc*PINK1 and hPINK1. We next explored whether Ins3, which is well conserved between *Tc*PINK1 and higher species, is critical for ubiquitin and Parkin Ubl recognition by hPINK1. Consistent with *Tc*PINK1, we observed abrogation of phosphorylation of ubiquitin and Parkin in cells expressing hPINK1 harbouring a deletion of Ins3 (Δ285–294 – corresponding to deletion in *Tc*PINK1 Δ261–270) in striking contrast to cells expressing hPINK1 containing a deletion in Ins1 (Δ180–209) or a deletion in the proximal region of Ins2 (Δ245–265 - corresponding to βi-strand and αi-helix of *Tc*PINK1). Overall, these findings indicate that hPINK1 and *Tc*PINK1 share similar mechanisms for substrate recognition and kinase activity, suggesting that the *Tc*PINK1 structure will provide an important framework to direct future studies aimed at structural studies of the hPINK1 enzyme.

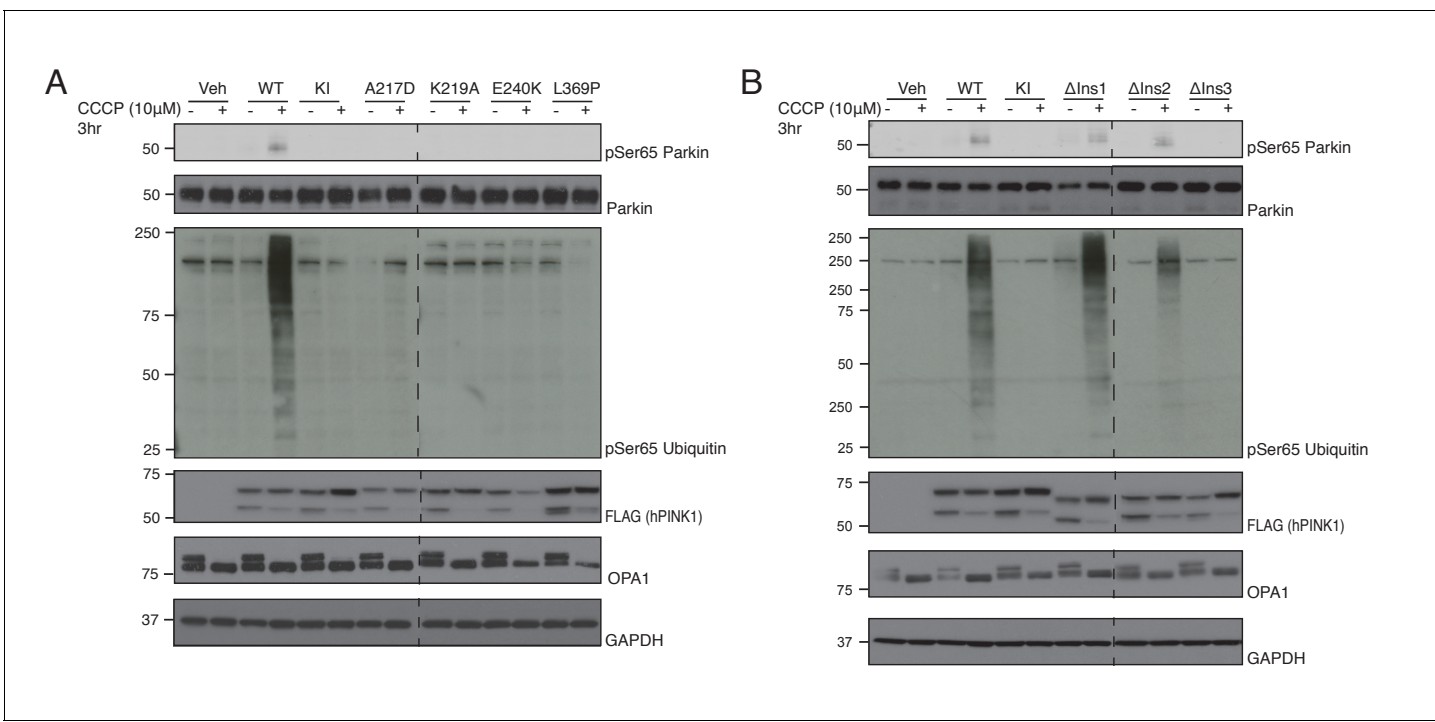

**Figure 4.** Conservation of ATP and substrate binding mechanism in hPINK1. (**A**) hPINK1 knock out HeLa cells transiently co-expressing WT human Parkin and 3xFLAG tagged hPINK1 WT, KI or hPINK1 ATP-binding mutants A217D, K219A, E240K, L369P. A217, K219, E240 and L369 correspond to A194, K196, E217 and L344 of *Tc*PINK1, respectively. Cells were stimulated with 10 µM CCCP or DMSO for 3 hr. Lysates were subjected to immunoblotting as follows: pSer65 Parkin (anti-phospho-Parkin); Parkin (anti-Parkin), pSer65 ubiquitin (anti-phospho-ubiquitin), hPINK1 (anti-FLAG), OPA1 (anti-OPA1) and GAPDH (anti-GAPDH). (**B**) hPINK1 knock out HeLa cells transiently co-expressing WT human Parkin and 3xFLAG tagged hPINK1 WT, KI or hPINK1 Ins mutants ΔIns1 (180–209 deletion), ΔIns2 (245–265 deletion), ΔIns3 (285-294). Cells were stimulated with 10 µM CCCP or DMSO for 3 hr. Lysates were subjected to immunoblotting as follows: pSer65 Parkin (anti-phospho-Parkin); Parkin (anti-Parkin), pSer65 ubiquitin (anti-phospho-ubiquitin), hPINK1 (anti-FLAG), OPA1 (anti-OPA1) and GAPDH (anti-GAPDH).
DOI: https://doi.org/10.7554/eLife.29985.009

The following figure supplement is available for figure 4:

**Figure supplement 1.** Sequence alignment of *Tc*PINK1 and hPINK1.
DOI: https://doi.org/10.7554/eLife.29985.010

## Molecular mechanisms of inactivating Parkinsonism mutations in hPINK1

Mutations in hPINK1 are a leading cause of familial early-onset PD and whilst previous studies have demonstrated that mutations disrupt hPINK1-dependent activity and mitophagy in cells (*McWilliams and Muqit, 2017*; *Deas et al., 2009*), very little is known about how these impact hPINK1 structure. Of the approximately thirty reported homozygous or compound heterozygous pathogenic mutations (*Deas et al., 2009*), twenty are conserved in *Tc*PINK1 (*Supplementary file 1*). Mapping of these pathogenic mutations onto the structure of *Tc*PINK1 (*Figure 5*, *Supplementary file 1*), reveals that they cluster within functionally distinct regions of hPINK1. A major group of PD-associated mutations lie within the ATP-binding pocket including Ala217Asp, Glu240Lys, Ala244Gly and Leu369Pro and perturb interactions with ATP. A second group of mutations that lie in the catalytic motif or activation loop include Gly386Ala, Pro416Arg/Leu and Glu417Gly (*Figure 5A,C*) result in catalytically inactive hPINK1 (*Deas et al., 2009*). The His271Gln mutation lies within the critical distal Ins2 loop of hPINK1 (*Figure 5A,C*), that anchors the proximal Ins2 to the C-lobe (*Figure 3C*), and has previously been found to abolish catalytic activity in vitro (*Woodroof et al., 2011*) as well as hPINK1 activity in cells (*Okatsu et al., 2012*). In contrast, the Gly309Asp (G309D) PD mutation located within Ins3 (*Figure 2*) would be predicted to perturb hPINK1 and substrate (ubiquitin/Ubl domain of Parkin) interaction as our analysis suggests that Ins3 is required for ubiquitin phosphorylation (*Figure 2C*, *Figure 2—figure supplement 1A*). Consistent with this, a previous study showed that the G309D hPINK1 mutant fails to phosphorylate Parkin Ubl in cells (*Iguchi et al., 2013*) whereas the equivalent *Tc*PINK1 G285D mutant continues to exhibit catalytic activity towards generic and peptide substrates in vitro (*Woodroof et al., 2011*). The remainder of mutations mainly affects residues critical for structural integrity of hPINK1. In particular, the CTE of *Tc*PINK1 forms a hydrophobic core against the E, G and H helices of the C-lobe (*Figure 5B*), and a cluster of PD mutations in hPINK1 including Leu539Phe, Leu347Pro and Leu489Pro located in this vicinity suggest the importance of this hydrophobic core (*Figure 5B*). Attempts to purify soluble protein for a number of point mutants targeting this hydrophobic interface were unsuccessful, underscoring the importance of this hydrophobic core which explains loss of kinase activity of *Tc*PINK1 upon CTE deletion in previous studies (*Woodroof et al., 2011*). Although it has previously been suggested that CTE may be important for dimerization (*Okatsu et al., 2013*), we did not find evidence for a functional dimer in our *Tc*PINK1 purification procedures or crystal structure.

Overall the *Tc*PINK1 structure provides molecular insights into the structural regulation of hPINK1 and in particular defines functional roles for its unique loop insertions in ubiquitin substrate recognition and catalytic activity. The structure provides clarity on the impact of loss of function disease-associated mutations, which may stimulate future drug discovery efforts for both familial and idiopathic PD.

## Materials and methods

### Protein expression, purification and crystallography

BL21 codon plus (DES) RIPL (Stratagene) were transformed with pET15 6HIS SUMO *Tc*PINK1$^{S150-D570\ \Delta I261-L270\ S205E\ E527A\ K528A}$ (available from MRC PPU Reagents and Services: http://mrcppureagents.dundee.ac.uk). For overexpression, 1 L of Terrific broth (TB) medium was inoculated with 10 ml of overnight culture. Cultures were propagated in a shaking incubator (Infors HT) at 37°C and 200 rpm until OD$_{600}$ ~0.6–0.8. Temperature was then reduced to 16°C upon which cultures were supplemented with 100 μM IPTG and grown overnight. Cells were harvested by centrifugation at 4000 rpm for 25 min (Beckman coulter J6-Mi centrifuge). Cell pellets were carefully resuspended in ~4 × cell volume of lysis buffer (50 mM EPPS pH 8.6, 500 mM NaCl, 5% glycerol, 3% sucrose, 5 mM imidazole) at 12°C by gentle shaking (150 rpm). Before lysis, 0.5 mM TCEP pH 7.5 (from Apollo Scientific, United Kingdom), 0.1 mM Pefabloc (Sigma Aldrich, United Kingdom), and 10 μM Leupeptin (Apollo Scientific) were added to the mixture. Cells were then lysed by sonication (6 × 10 s pulses at 10 s intervals, 45% amplitude) and to remove the cell debris, the cell lysates were centrifuged at 28,000 rpm for 25 min at 4°C (Beckman Avanti J-30 I centrifuge).

For protein purification, the supernatant was incubated with Nickel sepharose beads previously equilibrated with 10 CV binding buffer (50 mM EPPS pH 8.6, 500 mM NaCl, 5% glycerol, 3%

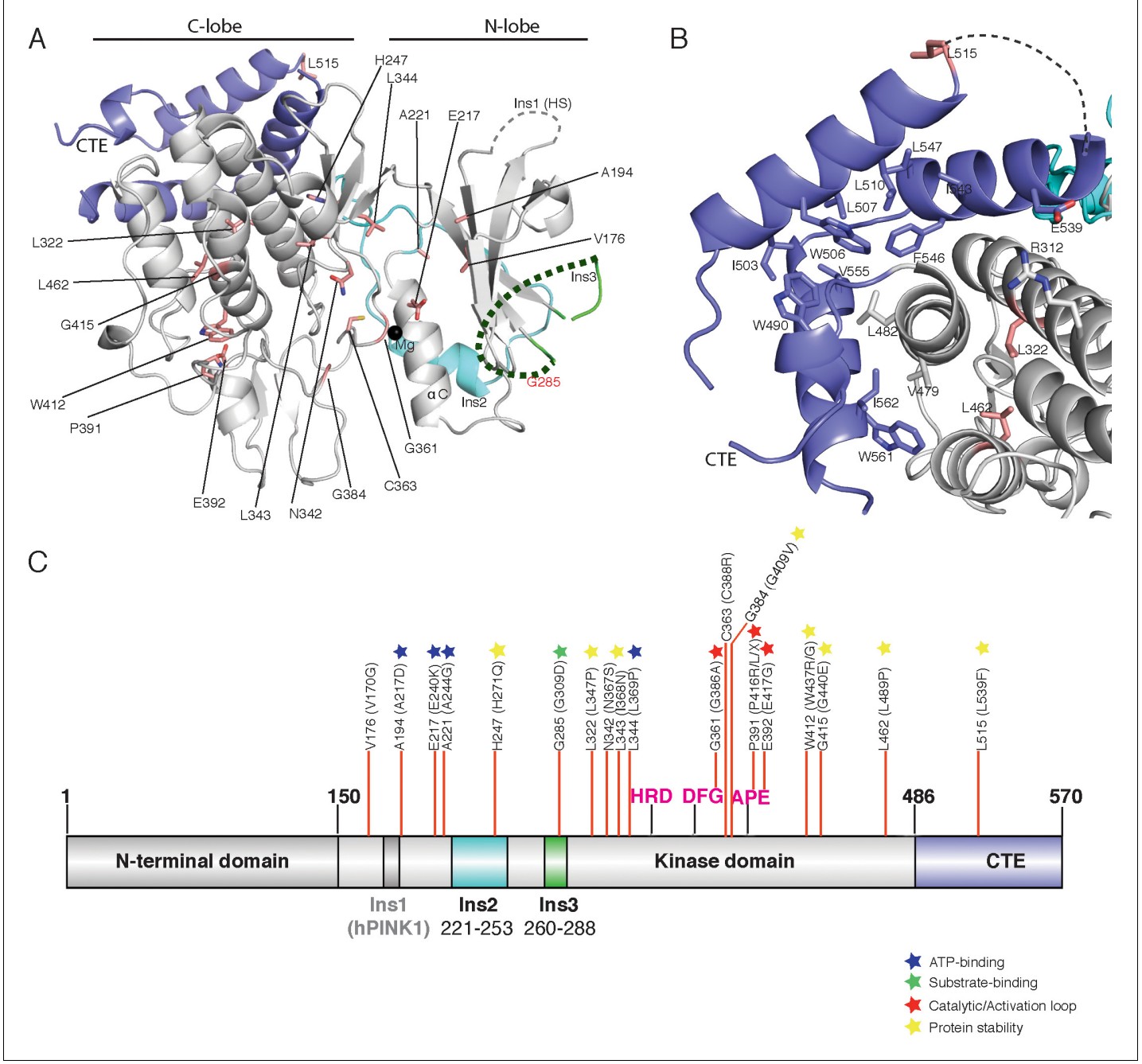

**Figure 5.** Spatial map of Parkinson's disease (PD) linked mutations within hPINK1. (**A**) Mapping of PD mutations onto the *Tc*PINK1 structure (coloured as in *Figure 1B*). Pathogenic hPINK1 disease mutant residues are shown in red sticks; disease mutants in the disordered region (represented by dashed line) of Ins3 are labelled in red characters. (**B**) Interface of the CTE and C-lobe of *Tc*PINK1 showing the hydrophobic core (shown in grey or blue) of the *Tc*PINK1 kinase domain. hPINK1 PD mutations in the vicinity of the hydrophobic core are shown in red sticks. (**C**) Location of hPINK1 PD mutations (marked with red lines) on the primary *Tc*PINK1 sequence. hPINK1 mutations are shown in parenthesis.

DOI: https://doi.org/10.7554/eLife.29985.011

sucrose, 5 mM imidazole, 0.5 mM TCEP, pH 7.5) for 1 hr at 4°C on a roller. Thereafter, the beads were collected by filtration and washed thoroughly (800 ml −1 L) with binding buffer containing 30 mM imidazole. The protein was then eluted from the beads using the elution buffer (50 mM EPPS pH 8.6, 500 mM NaCl, 5% glycerol, 3% sucrose, 400 mM imidazole, 0.5 mM TCEP, pH 7.5). The eluted protein was mixed with His-SENP1 protease (1 mg of His-SENP1 per 25 mg of *Tc*PINK1) to

cleave off the SUMO tag and dialyzed into 30 mM HEPES pH 7.5, 500 mM NaCl, 5% glycerol, 3% sucrose, 0.25 mM TCEP, pH 7.5 at 4°C overnight (using 3.5 kDa SnakeSkin dialysis tubing; Thermo Fisher). The dialyzed mixture was subjected to negative pull down using Nickel sepharose resin to remove the His-SUMO and TcPINK1 was concentrated to ~5 ml. Reductive methylation of lysine residues was then performed (as described in Hampton Research Protocols) before loading onto a gel filtration column (HiLoad Superdex 16/600, GE Healthcare), previously equilibrated with the gel filtration buffer (25 mM HEPES, pH 7.5, 300 mM NaCl, 5% glycerol, 3% sucrose, 2 mM TCEP, pH 7.5). Following gel filtration, methylated TcPINK1 was concentrated to 9 mg/ml flash frozen in liquid nitrogen and stored at −80°C for further use. For overexpression, purification and methylation of selenomethionine (SeMet) TcPINK1 derivatives, an identical protocol was used (using MD12-500 kit from Molecular Dimensions to incorporate selenomethionine). Native and selenomethionine derivatised TcPINK1 was crystallised using 5 mg/ml of protein mixing with 200 mM MgCl$_2$, 100 mM HEPES pH 7.0, 20% PEG 6000 and 200 mM MgCl$_2$ or 100 mM HEPES pH 7.5, 25% PEG 3350 and 200 mM MgCl$_2$, respectively, under vapor diffusion sitting drop method. Crystals were flash frozen using 20% of glycerol with mother liquor. Data were collected at ESRF, with a SeMet anomalous data set collected at the selenium absorption edge. Selenomethionine-incorporated and native crystals diffracted to 3.4 Å and 2.8 Å, respectively. Data were processed using XDS (*Kabsch, 2014*) and AIMLESS (*Winn et al., 2011*). SAD phasing was performed on the anomalous data set using CRANK2 in the CCP4 program suite. Phases were improved by iterative cycles of manual model building and refinement using Coot (*Emsley and Cowtan, 2004*) and Refmac (*Winn et al., 2011*), respectively (*Table 1*). Ramachandran values were calculated using MolProbity (*Chen et al., 2010*) with 94% of residues in the most favored regions.

## Protein purification from *E. coli* for kinase assays

Full length wild-type and mutant TcPINK1 were expressed in *E. coli* (BL21-codonplus) as maltose-binding protein fusion proteins. Cells were grown at 37°C, induced with 250 μM IPTG at OD$_{600}$ 0.6 and were further grown at 16°C for 16 hr. Cells were pelleted at 4000 r.p.m., and then lysed by sonication in lysis buffer. Lysates were clarified by centrifugation at 30 000 *g* for 30 min at 4°C followed by incubation with 1 ml per litre of culture of amylose resin for 1.5 hr at 4°C. The resin was washed thoroughly in wash buffer, then equilibration buffer, and proteins were then eluted. Proteins were dialysed overnight at 4°C into storage buffer, snap-frozen and stored at −80°C until use. Kinase assay substrates (ubiquitin, Parkin Ubl, and TcPINK1 125-end D359A) were expressed as 6His-SUMO fusion proteins using the same conditions. Cleared lysate was incubated with 1 ml Ni$^{2+}$-NTA resin/litre of culture for 1.5 hr at 4°C. The resin was washed thoroughly in wash buffer, then equilibration buffer, and proteins were then eluted. Proteins were dialysed overnight at 4°C into storage buffer. 6His-SUMO tags were cleaved by the addition of SENP1 at a ratio of 1:10 protease:tagged-protein. Tag-cleaved protein was purified by thorough incubation with Ni$^{2+}$-NTA resin, and then snap-frozen and stored at −80°C until further use.

## Kinase assays

Reactions were set up in a volume of 40 μl, using 2 μg/30 ng of *E. coli*-expressed TcPINK1 and 2 μM of substrate, in 50 mM Tris–HCl (pH 7.5), 0.1 mM EGTA, 10 mM MgCl$_2$, 2 mM DTT and 0.1 mM [γ-$^{32}$P] ATP (approx. 500 cpm pmol$^{-1}$). Assays were incubated at 30°C with shaking at 1050 r.p.m. and terminated after 10 min by addition of SDS sample loading buffer. The reaction mixtures were then resolved by SDS–PAGE. Proteins were detected by Coomassie staining, and gels were imaged using an Epson scanner and dried completely using a gel dryer (Bio-Rad). Incorporation of [γ-$^{32}$P] ATP into substrates was analysed by autoradiography using Amersham hyperfilm. Quantification of [γ-$^{32}$P] incorporation into substrates was performed by Cerenkov counting of respective SDS-PAGE gel bands.

## Buffers for *E. coli* protein purification for activity assays

TcPINK1 and indicated mutants were expressed as maltose-binding protein fusion proteins: lysis buffer contained 50 mM Tris–HCl (pH 7.5), 150 mM NaCl, 1 mM EDTA, 1 mM EGTA, 5% (v/v) glycerol, 1% (v/v) Triton X-100, 0.1% (v/v) 2-mercaptoethanol, 1 mM benzamidine and 0.1 mM PMSF. Wash buffer contained 50 mM Tris–HCl (pH 7.5), 500 mM NaCl, 0.1 mM EGTA, 5% (v/v) glycerol,

**Table 1.** Diffraction data and refinement statistics.

| | TcPINK1 (Native) |
|---|---|
| Data collection | |
| Space group | P1 2$_1$ 1 |
| Cell dimensions | |
| a, b, c (Å) | 84.92 116.74 179.34 |
| α, b, g (°) | 90.00 94.29 90.00 |
| Wavelength (Å) | 1.03 |
| Resolution (Å) | 178.84–2.78 (2.83–2.78)* |
| $R_{merge}$ | 8.5 (122.4) |
| $I/\sigma(I)$ | 13.0 (1.4) |
| $CC_{1/2}$ | 99.9 (59.4) |
| Completeness (%) | 99.9 (99.8) |
| Redundancy | 6.7 (6.8) |
| Refinement | |
| Resolution (Å) | 178.84 (2.78) |
| No. reflections | 83545 |
| $R_{work}/R_{free}$ | 20.6/24.5 |
| No. atoms | |
| Protein | 16722 |
| Ligand/ion | 6 |
| Water | 30 |
| B factors | |
| Protein | 80.8 |
| Ligand/ion | 81.6 |
| Water | 70.9 |
| R.m.s. deviations | |
| Bond lengths (Å) | 0.01 |
| Bond angles (°) | 1.8 |

Single crystals were used for structure determination.

*Values in parentheses are for highest-resolution shell.

DOI: https://doi.org/10.7554/eLife.29985.012

0.03% (v/v) Brij-35, 0.1% (v/v) 2-mercaptoethanol, 1 mM benzamidine and 0.1 mM PMSF. Equilibration buffer contained 50 mM Tris–HCl (pH 7.5), 150 mM NaCl, 0.1 mM EGTA, 5% (v/v) glycerol, 0.03% (v/v) Brij-35, 0.1% (v/v) 2-mercaptoethanol, 1 mM benzamidine and 0.1 mM PMSF. Elution buffer was equilibration buffer with the addition of 12 mM maltose. Storage buffer was equilibration buffer with the addition of 0.27 M sucrose, and glycerol, PMSF and benzamidine were omitted.

Ubiquitin K63 tetramer was expressed as previously described (*Kristariyanto et al., 2015*). Lysis buffer contained 50 mM Tris–HCl (pH 7.5), 150 mM NaCl, 5% (v/v) glycerol, 20 mM imidazole, 1% (v/v) Triton X-100, 0.1% (v/v) 2-mercaptoethanol, 1 mM benzamidine and 0.1 mM PMSF. Wash buffer contained 50 mM Tris–HCl (pH 7.5), 500 mM NaCl, 5% (v/v) glycerol, 20 mM imidazole, 0.03% (v/v) Brij-35, 0.1% (v/v) 2-mercaptoethanol, 1 mM benzamidine and 0.1 mM PMSF. Equilibration buffer contained 50 mM Tris–HCl (pH 7.5), 150 mM NaCl, 5% (v/v) glycerol, 0.03% (v/v) Brij-35, 0.1% (v/v) 2-mercaptoethanol, 1 mM benzamidine and 0.1 mM PMSF. Elution buffer was equilibration buffer with the addition of 200 mM imidazole. Storage buffer was equilibration buffer with the addition of 0.27 M sucrose, and glycerol–PMSF and benzamidine were omitted.

## Cell culture

HeLa hPINK1 knockout cell lines were obtained from Richard Youle (NIH) and were cultured using DMEM (Dulbecco's modified Eagle's medium) supplemented with 10% FBS (foetal bovine serum), 2 mM L-glutamine, 100 U/mL penicillin and 0.1 mg/mL streptomycin. Cells were transiently transfected with WT Parkin and 3xFLAG tagged PINK1 using polyethylene method. Mitochondria were uncoupled using 10 μM CCCP (Sigma) dissolved in DMSO for 3 hr. 1% Triton solubilized lysates were resolved on SDS-PAGE, and wet transferred onto nitrocellulose membranes. Membranes were blocked with 5% BSA followed by overnight incubation with the indicated antibodies in 5% BSA or milk in TBST at four degrees. Membranes were incubated with HRP-conjugated secondary antibodies diluted (1: 10,000) in 1X TBST with 5% BSA for 1 hr at room temperature followed by exposure with ECL, pSer65 Parkin membranes were incubated with secondary antibodies conjugated with LI-COR IRDye at 1:10,000 dilution in 1X TBST with 5% BSA. List of antibodies used in the present study are mentioned in *Supplementary file 2*.

## Acknowledgements

This work was funded by a Wellcome Trust Senior Fellowship to MMKM. (101022/Z/13/Z), the Medical Research Council and a joint MMKM and DMFvA grant from Parkinson's UK (G-1506; Fife and Ayrshire Parkinson's UK Branches). DMFvA is funded by a Wellcome Trust Investigator Award (110061). We thank Prosenjit Pal for assistance in identifying PINK1 disease mutants. We thank the European Synchrotron Radiation Facility for time on beamline ID23-1. Coordinates and structure factors have been deposited with the PDB database (PDB ID 5OAT). We thank Dr. Yogesh Kulathu & Prof. Helen Walden for useful discussions. We also thank excellent technical support provided by MRC Protein Phosphorylation and Ubiquitylation Unit (PPU) DNA sequencing service (coordinated by Nicholas Helps), the MRC PPU tissue culture team (coordinated by Laura Fin), the MRC PPU reagents and services team (coordinated by Hilary McLauchlan and James Hastie), and Axel Knebel. We also thank Mark Dorward for preparing crystal screening plates.

## Additional information

### Funding

| Funder | Grant reference number | Author |
|---|---|---|
| Medical Research Council | | Andrew D Waddell |
| Biotechnology and Biological Sciences Research Council | Council (BBSRC) [grant number BB/M010996/1] | Andrew M Shaw |
| Parkinson's UK | G-1506 | Miratul MK Muqit<br>Daan MF van Aalten |
| Wellcome | 101022/Z/13/Z | Miratul MK Muqit |
| Wellcome | 110061 | Daan MF van Aalten |

The funders had no role in study design, data collection and interpretation, or the decision to submit the work for publication.

### Author contributions

Atul Kumar, Software, Formal analysis, Validation, Investigation, Methodology, Writing—original draft, Writing—review and editing; Jevgenia Tamjar, Resources, Validation, Investigation, Methodology; Andrew D Waddell, Resources, Validation, Investigation, Visualization, Methodology; Helen I Woodroof, Resources, Software, Investigation, Methodology; Olawale G Raimi, Andrew M Shaw, Resources, Investigation, Methodology; Mark Peggie, Resources; Miratul MK Muqit, Conceptualization, Supervision, Funding acquisition, Validation, Visualization, Writing—original draft, Project administration, Writing—review and editing; Daan MF van Aalten, Conceptualization, Resources, Formal analysis, Supervision, Funding acquisition, Validation, Visualization, Writing—original draft, Project administration, Writing—review and editing

## Author ORCIDs

Atul Kumar ⓘ https://orcid.org/0000-0001-6839-2299
Olawale G Raimi ⓘ http://orcid.org/0000-0003-2782-7318
Miratul MK Muqit ⓘ https://orcid.org/0000-0001-9733-2404
Daan MF van Aalten ⓘ https://orcid.org/0000-0002-1499-6908

## Decision letter and Author response

Decision letter https://doi.org/10.7554/eLife.29985.018
Author response https://doi.org/10.7554/eLife.29985.019

# Additional files

## Supplementary files

• Supplementary file 1. List of hPINK1 PD mutations conserved in *Tc*PINK1 and their predicted impact
DOI: https://doi.org/10.7554/eLife.29985.013

• Supplementary file 2. List of Antibodies used and their source
DOI: https://doi.org/10.7554/eLife.29985.014

• Transparent reporting form
DOI: https://doi.org/10.7554/eLife.29985.015

## Major datasets

The following dataset was generated:

| Author(s) | Year | Dataset title | Dataset URL | Database, license, and accessibility information |
|---|---|---|---|---|
| Kumar A, Tamjar J, Waddell AD, Woodroof HI, Raimi OG, Shaw AM, Peggie M, Muqit MMK, van Aalten DMF | 2017 | PINK1 structure | https://www.rcsb.org/pdb/explore.do?structureId=5OAT | 5OAT |

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
