## [Decision Letter]

Thank you for submitting your article "Structure and activation mechanism of the Parkinson's disease kinase PINK1" for consideration by *eLife*. Your article has been reviewed by three peer reviewers, and the evaluation has been overseen by Tony Hunter as the Senior and Reviewing Editor. The following individuals involved in review of your submission have agreed to reveal their identity: Kalle Gehring (Reviewer #1); Stefan Knapp (Reviewer #2).

The reviewers have discussed the reviews with one another and the Reviewing Editor has drafted this decision to help you prepare a revised submission.

The reviewers all found this description of a first structure of a member of the PINK1 family of kinases to represent an exciting advance in the field, but had a number of suggestions for improvement and reorganization.

The main points to be taken into account in a revised version are as follows:

1) You need to stress in the title, Abstract and throughout the paper that the structure is that of TcPINK1. Otherwise, the reader may be led to believe that the paper describes the structure of the human PINK1 catalytic domain. In addition, since you have not established the mechanism of activation of human PINK1, "activation mechanism" should be removed from the title.

2) Since TcPINK1 is constitutively active, you should be more cautious about what this catalytic domain structure tells us about the activation mechanism of human PINK1, i.e. you need to state/discuss more explicitly what regulatory mechanisms derived from the structure of TcPINK1 are relevant to hPINK1 and what aspects of hPINK1 regulation might be different.

3) Your proposal that Ins2 activates the catalytic domain through a cyclin-like interaction with the C-helix is intriguing and novel, but, although you carried out mutagenesis experiments to define the ATP-binding pocket, you did not conduct any direct mutagenesis experiments to verify that an interaction between Ins2 and the C-helix is important for activity of the catalytic domain. As you indicate, you have previously reported that the H271Q mutation, which lies in Ins2, reduces TcPINK1 kinase activity (Woodroof et al., 2011), but there is no discussion of why this mutation should be deleterious, and the H271 residue and its contacts are not shown in Figure 4. Ideally, you need to analyze the effects of mutating a C-helix interface residue in Ins2 predicted by your structure to be inactivating to bolster the conclusion that Ins2 activates TcPINK1 by a cyclin-like contact with the C-helix.

4) Both human and TcPINK1 have been shown to be activated by phosphorylation of Ser228 (human)/Ser205 (TcPINK1), but this site is not discussed at all. Instead, you describe the contacts and supposed regulatory importance of the Ins2 region but with no indication of how those contacts would be modified as part of a regulatory mechanism. The argument in the Results and Discussion that human PINK1 is regulated and TcPINK1 constitutive is unproven. An equally plausible explanation is that the human protein is unstable when purified and/or inherently less active. Related to this, the statement that the authors were "we were unable to confirm […] by mutagenesis" the significance of the amino acid differences between human and TcPINK1 (subsection “Mechanisms of PINK1 autoactivation”) needs to be better explained. Was it a negative result or was the experiment unsuccessful?

*eLife* revise decision letters normally do not provide the full reviews, but only a summary of the main issues that need to be addressed. However, in this case, we have decided to provide the three reviews in full so that you can see all the concerns, and revise the paper accordingly.

*Reviewer 1:*

Kumar and colleagues report the structure of PINK1, a kinase involved in mitochondrial autophagy and a recessive early-onset form of Parkinson's disease. The discovery three years ago of phosphorylation of ubiquitin by PINK1 has generated intense interest in the kinase and its potential as a drug target. The authors have determined the structure of an insect orthologue and modeled its interaction with its substrates, ubiquitin and ATP. Using the structure, they have generated a mutant that is defective for ubiquitin phosphorylation but retains activity for autophosphorylation and myelin basic protein.

The main problem with the manuscript is its presentation. The figures are poorly conceived and repetitive and the order of presentation of the results is confusing. From a scientific standpoint, the article is outstanding and a significant, major contribution to the field.

1) The presentation of the structure in Figure 1 will be more easily understood by the large kinase community if the standard orientation is used: the N-lobe above the C-lobe and the activation loop toward the reader. The activation loop should be indicated in the figure and the model of ATP binding derived from the LIM kinase structure removed. I suggest moving Figure 2 to Figure 1. While I understand the authors' motivation for showing the ATP in Figure 1, it is a bit confusing since i) it is a model and not part of the crystal structure determined and ii) the DALI search that identifies the LIM kinase structure isn't presented until much later in the paper. To add to the confusion, the figure does not show AMP-PNP. Instead, there appears to be a sulfur on the γ phosphate. (As an aside, I'll point out that given that it is a model, there is no reason to show AMP-PNP rather than ATP. The authors should also make sure that the font sizes are consistent across the different panels and aim to keep the figures as simple and easy to interpret as possible.)

2) As an alternative to the current organization, the authors might consider putting all of the enzymatic kinase assays into a single figure. My suggestion would be the structure in Figure 1 with location of PD mutations, Figure 3—figure supplement 1, and possibly the modeled ATP. Figure 2 could be the kinase assays and Figure 3 could present the comparison with LIMK1 and docking of ubiquitin. This should help to remove some of the duplications. For example, Figure 1 and Figure 4 would be combined and only shown once.

3) Move Table 1 to the supplemental material and Figure S3 should be part of the kinase activity figure in the main manuscript.

4) Add paragraph breaks to the Introduction to improve the legibility.

5) The text in the Results and Discussion should explain the use S205E phosphomimetic referencing the Woodroof 2011 paper. I found it curious that there was no discussion of the importance of this residue for kinase activity when it reportedly has a greater effect on activity than loss of S207. E205 is shown in the example electron density but no reference is made to its interactions with other residues. S207 and E205 might be hidden somewhere in Figure 1/2 but I couldn't find them.

6) The explanation of the activation of PINK1 in the subsection “Mechanisms of PINK1 autoactivation” was confusing. Since the TcPINK1 kinase domain adopts an active conformation, it wasn't clear how to interpret the intermolecular interactions listed. Do the authors expect that these will be lost in human PINK1? If so, how is human PINK1 activated? The kinase assays show TcPINK1 autophosphorylates. Where is the phosphorylation site? Does this regulate its activity or is TcPINK1 constitutively active?

7) The authors should expand the discussion of the relationship of PINK1 to other kinases. Sequence analysis (BLAST) shows much greater similarity to calmodulin kinases than LIMK1 so the observation of LIMK1 as the top hit is unexpected. What were the other DALI hits? Is LIMK1 really the best model for positioning of the ATP?

8) In the figure legend for Figure 4, it isn't clear which are confirmed effects of the mutations and which are predicted.

*Reviewer 2:*

The paper by Kumar et al. entitled "Structure and activation mechanism of the Parkinson's disease kinase PINK1"reports the crystal structure of the catalytic and C-terminal domain of *Tribolium castaneum* homologue of the kinase PINK1. The structure revealed a canonical catalytic domain fold with a helical C-terminal domain. From the three inserts present in human PINK1 the first insert is missing in TcPINK1, the second insert was deleted and the third insert was present but unstructured. The authors report that similar to the cyclin interaction in CDKs an α helical motif stabilizes the active state. In addition, the disordered third insert motif seems to be important for substrate recognition (ubiquitin) and a molecular model of the PINK1 substrate interaction was constructed based on the recently published LIMK substrate (cofilin) complex. Finally the authors analyze the effect of mutations found in PD revealing that many mutations are in positions that are important for PINK1 catalytic function such as the ATP binding pocket.

Human PINK1 is a structurally very diverse kinase with a strong link to the development of PD. To date structural studies on the human enzyme have been hampered due to difficulties expressing stable and large amounts of this unusual kinase. A structural model that provides insights into the regulation of PINK1, the role of its unusual inserts and the consequences of PD mutations is therefore very welcome and informative. However, I had to read until the end of the Introduction to find out that the presented structure is actually based on the TcPINK1 sequence. I think therefore that this should be highlighted in the title, Abstract and impact statement. Also a more detailed sequence comparison would be helpful to judge how this model can help to provide insight on the regulation of the human enzyme (maybe as supplemental material). How high is the overall sequence homology? In addition, the deletion of loop regions and mutations aiding crystallization should be annotated in Figure 1.

It seems that there are clear differences in the activation of TcPINK and human PINK1. For instance the first large insert is missing. Is this insert required for human PINK1 function? In addition, TcPINK1 is constitutively active whereas the human enzyme may require activation. While certain aspects of PINK1 regulation (e.g. the substrate recognition) may be shared between TcPINK1 and hPINK1 there seems to be clear differences in the regulation of this enzyme in higher eukaryotes. Despite these limitations I think however that the presented structural data are very useful and informative and based on the interested in the role of PINK1 in PD the article may trigger functional studies on the human enzyme and it will be widely read. My recommendation is therefore to simply make the paper more transparent and to highlight the limitation of the current structural model more clearly in the revised version.

*Reviewer 3:*

This manuscript reports the crystal structure of PINK1, a Ser/Thr protein kinase involved in regulation (phosphorylation) of ubiquitin. The authors crystallized a *T. castaneum* orthologue of human PINK1 (TcPINK1 and hPINK1), the construct of which included, in addition to the kinase domain, N-terminal domains Ins2 and Ins3 and C-terminal domain CTE. The structure was determined at 2.8 Å resolution (not 2.7 Å, as reported in Results) and reveals the structure of the kinase domain, Ins2, and the CTE. The kinase domain is observed to be in an active state, with the alphaC-beta3 salt bridge intact and a DFG-in activation loop configuration. Ins2 and the CTE make interactions with the kinase domain that are probably important for kinase regulation.

Major concern:

Contrary to their claim in the Title and Abstract, the authors do not really reveal the activation mechanisms of hPINK1. They state that TcPINK1 is active in vitro (the crystal structure evidently shows why), but hPINK1 is not active in vitro. Why hPINK1 is not active and how it becomes activated are not explored.

Other substantive concerns:

1) The authors speculate from the TcPINK1 crystal structure that Ins2 activates the kinase domain via a cyclin-CDK mechanism (involving αC). While this seems quite plausible based on the structure (and is arguably the most important feature of the structure), the authors did not perform any mutagenesis experiments to verify this.

2) The authors speculate that Ins3, which is just upstream of the kinase domain and which is disordered in their crystal structure, interacts with ubiquitin to provide substrate specificity. To test this, they make a 10-residue deletion construct of Ins3 in TcPINK1 and show that ubiquitin phosphorylation is abrogated, whereas MBP phosphorylation (non-specific substrate, negative control) is less affected. As shown in Figure 3, though, MBP phosphorylation is down over 5-fold, indicating that the Ins3 deletion is probably affecting PINK1 stability, in addition to affecting (possibly) specific ubiquitin phosphorylation. A generic peptide substrate might serve as a better negative control.

---

## [Author Response]

The reviewers all found this description of a first structure of a member of the PINK1 family of kinases to represent an exciting advance in the field, but had a number of suggestions for improvement and reorganization.The main points to be taken into account in a revised version are as follows:1) You need to stress in the title, Abstract and throughout the paper that the structure is that of TcPINK1. Otherwise, the reader may be led to believe that the paper describes the structure of the human PINK1 catalytic domain. In addition, since you have not established the mechanism of activation of human PINK1, "activation mechanism" should be removed from the title.2) Since TcPINK1 is constitutively active, you should be more cautious about what this catalytic domain structure tells us about the activation mechanism of human PINK1, i.e. you need to state/discuss more explicitly what regulatory mechanisms derived from the structure of TcPINK1 are relevant to hPINK1 and what aspects of hPINK1 regulation might be different.

We thank reviewers/editors for pointing this out. Exploiting the TcPINK1 structure, we have now included cell based experimental analysis of hPINK1 to show that the ATP-binding and ubiquitin substrate recognition mechanisms are conserved between hPINK1 and TcPINK1 (Figure 4). Employing cell based assays of hPINK1 we show that Ins1 (that is not conserved in TcPINK1) does not play any role in the regulation of hPINK1-directed phosphorylation of substrates (Figure 4).

We have included new functional data showing the importance of the basic patch of residues around the catalytic pocket of TcPINK1 for activation (Figure 3). These residues are not conserved in hPINK1 and we have speculatively engineered a TcPINK1-like basic patch into hPINK1, although this does not appear to lead to an increase of hPINK1 kinase activity (Author response image 1). As there may be technical reasons for the lack of activity particularly with stability and folding of recombinant hPINK1 (that was also raised by reviewers), we have not included these data in the main manuscript.

As requested we have removed "activation mechanism" from the title and Results. The new title is “Structure of PINK1 and mechanism of Parkinson’s disease associated mutations”.

We have now amended the Abstract to clearly state that the structure is of TcPINK1 to avoid any confusion.

**Author response image 1. respfig1:** Incorporation of basic residues around the catalytic pocket, identified from TcPINK1 structure (Figure 3), do not result in an active hPINK1 enzyme.

3) Your proposal that Ins2 activates the catalytic domain through a cyclin-like interaction with the C-helix is intriguing and novel, but, although you carried out mutagenesis experiments to define the ATP-binding pocket, you did not conduct any direct mutagenesis experiments to verify that an interaction between Ins2 and the C-helix is important for activity of the catalytic domain. As you indicate, you have previously reported that the H271Q mutation, which lies in Ins2, reduces TcPINK1 kinase activity (Woodroof et al., 2011), but there is no discussion of why this mutation should be deleterious, and the H271 residue and its contacts are not shown in Figure 4. Ideally, you need to analyze the effects of mutating a C-helix interface residue in Ins2 predicted by your structure to be inactivating to bolster the conclusion that Ins2 activates TcPINK1 by a cyclin-like contact with the C-helix.

We now demonstrate interactions between H247 (H271 in hPINK1) and the C-terminal extension (CTE). We further highlight that H247 lies within a highly conserved C-terminal loop of Ins2 (residues 243-253) (Figure 3, Figure 3—figure supplement 1). Mutational analysis of distinct truncations of Ins2 indicate that this C-terminal loop (243-253) of Ins2, where H247 resides, is crucial for PINK1 activity (Figure 3, Figure 3—figure supplement 1). Interestingly we find that deletion of the αi-helix (231-242) alone or the combined βi-strand and αi-helix (222-242) region of Ins2 does not significantly affect the kinase activity of TcPINK1 (Figure 3) or hPINK1 (Figure 4) under the assay conditions used. Therefore, the role of the proximal region of Ins2 remains uncertain and requires further investigation. We have substantially revised the text to explain these findings.

4) Both human and TcPINK1 have been shown to be activated by phosphorylation of Ser228 (human)/Ser205 (TcPINK1), but this site is not discussed at all. Instead, you describe the contacts and supposed regulatory importance of the Ins2 region but with no indication of how those contacts would be modified as part of a regulatory mechanism. The argument in the Results and Discussion that human PINK1 is regulated and TcPINK1 constitutive is unproven. An equally plausible explanation is that the human protein is unstable when purified and/or inherently less active. Related to this, the statement that the authors were "we were unable to confirm […] by mutagenesis" the significance of the amino acid differences between human and TcPINK1 (subsection “Mechanisms of PINK1 autoactivation”) needs to be better explained. Was it a negative result or was the experiment unsuccessful?

We thank the reviewers and editor for directing us to further investigate and discuss the role of Ser205 phosphorylation. We now include new data on Ser205 that has revealed an exciting and functionally important contribution of Ser205 phosphorylation in PINK1 activity towards substrates. Structural analysis indicates that Ser205 phosphorylation forms a bowl with Ins3 (Figure 2), which plays a crucial role in ubiquitin recognition and phosphorylation (Figure 2 and Figure 2—figure supplement 1). We have also studied the role of another phosphorylation site, Ser207, that was captured in the crystal structure, but this lies on the surface and apparently does not affect PINK1 phosphorylation of ubiquitin (Figure 2 and Figure 2—figure supplement 1).

We have included a detailed sequence alignment between TcPINK1 and hPINK1 (Figure 4—figure supplement 1), alongside our previous multiple sequence alignment between various organism highlighting important regions around the αC-helix, Ins2 and the activation loop (Figure 3—figure supplement 1).

Regarding the basic patch residues within the catalytic pocket of TcPINK1, we have now explained the differences between hPINK1 and TcPINK1 as follows “To explore which residues were critical for PINK1 catalysed phosphorylation of its substrates we investigated point mutants of residues lying within this region. […] Furthermore, Arg240 and Arg241 of *Tc*PINK1 are replaced by Gly264 and Pro265 in *h*PINK1, respectively (Figure 3—figure supplement 1).”

We hypothesized that these basic residues differences could explain the lack of recombinant hPINK1 activity and have addressed this experimentally as detailed in our response above to points 1 and 2.

We agree with the reviewer that an “equally plausible explanation is that the human protein is unstable when purified and/or inherently less active”. Therefore, we have amended the manuscript to remove references to the activation mechanism of hPINK1.

eLife revise decision letters normally do not provide the full reviews, but only a summary of the main issues that need to be addressed. However, in this case, we have decided to provide the three reviews in full so that you can see all the concerns, and revise the paper accordingly.Reviewer 1:[…] The main problem with the manuscript is its presentation. The figures are poorly conceived and repetitive and the order of presentation of the results is confusing. From a scientific standpoint, the article is outstanding and a significant, major contribution to the field.1) The presentation of the structure in Figure 1 will be more easily understood by the large kinase community if the standard orientation is used: the N-lobe above the C-lobe and the activation loop toward the reader. The activation loop should be indicated in the figure and the model of ATP binding derived from the LIM kinase structure removed. I suggest moving Figure 2 to Figure 1. While I understand the authors' motivation for showing the ATP in Figure 1, it is a bit confusing since i) it is a model and not part of the crystal structure determined and ii) the DALI search that identifies the LIM kinase structure isn't presented until much later in the paper. To add to the confusion, the figure does not show AMP-PNP. Instead, there appears to be a sulfur on the γ phosphate. (As an aside, I'll point out that given that it is a model, there is no reason to show AMP-PNP rather than ATP. The authors should also make sure that the font sizes are consistent across the different panels and aim to keep the figures as simple and easy to interpret as possible.)2) As an alternative to the current organization, the authors might consider putting all of the enzymatic kinase assays into a single figure. My suggestion would be the structure in Figure 1 with location of PD mutations, Figure 3—figure supplement 1, and possibly the modeled ATP. Figure 2 could be the kinase assays and Figure 3 could present the comparison with LIMK1 and docking of ubiquitin. This should help to remove some of the duplications. For example, Figure 1 and Figure 4 would be combined and only shown once.3) Move Table 1 to the supplemental material and Figure S3 should be part of the kinase activity figure in the main manuscript.4) Add paragraph breaks to the Introduction to improve the legibility.

We have now modelled ATP-γ-S instead of AMP-PNP. We have also made all other necessary changes to improve the presentation quality. However, we have refrained from grouping all the mutational assay results together and instead placed this next to the structural data as we think this will allow readers to understand the appropriate context of the assay data.

5) The text in the Results and Discussion should explain the use S205E phosphomimetic referencing the Woodroof 2011 paper. I found it curious that there was no discussion of the importance of this residue for kinase activity when it reportedly has a greater effect on activity than loss of S207. E205 is shown in the example electron density but no reference is made to its interactions with other residues. S207 and E205 might be hidden somewhere in Figure 1/2 but I couldn't find them.6) The explanation of the activation of PINK1 in the subsection “Mechanisms of PINK1 autoactivation” was confusing. Since the TcPINK1 kinase domain adopts an active conformation, it wasn't clear how to interpret the intermolecular interactions listed. Do the authors expect that these will be lost in human PINK1? If so, how is human PINK1 activated? The kinase assays show TcPINK1 autophosphorylates. Where is the phosphorylation site? Does this regulate its activity or is TcPINK1 constitutively active?

We have addressed these points separately above in the response to the Editor’s comments (comments 1-4).

7) The authors should expand the discussion of the relationship of PINK1 to other kinases. Sequence analysis (BLAST) shows much greater similarity to calmodulin kinases than LIMK1 so the observation of LIMK1 as the top hit is unexpected. What were the other DALI hits? Is LIMK1 really the best model for positioning of the ATP?

We appreciate that calmodulin kinase is more identical to PINK1 at the *sequence* level, however, according to Z score in DALI, LIM Kinase is the most *structurally* similar kinase, please see the details of DALI search in Author response image 2.

**Author response image 2. respfig2:** 

8) In the figure legend for Figure 4, it isn't clear which are confirmed effects of the mutations and which are predicted.

In Figure 4 (Figure 5 in the revised manuscript), the predicted effects are on the basis of the region in the structure where the mutation is located; and the functional role of that region e.g. activation loop, catalytic motif, ATP binding pocket, substrate binding pocket, Ins3 or hydrophobic core of the protein.

Reviewer 2:[…] Human PINK1 is a structurally very diverse kinase with a strong link to the development of PD. To date structural studies on the human enzyme have been hampered due to difficulties expressing stable and large amounts of this unusual kinase. A structural model that provides insights into the regulation of PINK1, the role of its unusual inserts and the consequences of PD mutations is therefore very welcome and informative. However, I had to read until the end of the Introduction to find out that the presented structure is actually based on the TcPINK1 sequence. I think therefore that this should be highlighted in the title, Abstract and impact statement. Also a more detailed sequence comparison would be helpful to judge how this model can help to provide insight on the regulation of the human enzyme (maybe as supplemental material). How high is the overall sequence homology? In addition, the deletion of loop regions and mutations aiding crystallization should be annotated in Figure 1.

We have addressed these points separately above in the response to the Editor’s comments (comments 1-4) including change of title, highlighting that the study is based upon TcPINK1 in the Abstract, inclusion of detailed sequence comparison (Figure 4—figure supplement 1) and describing the crystallisation construct in Figure 1.

It seems that there are clear differences in the activation of TcPINK and human PINK1. For instance the first large insert is missing. Is this insert required for human PINK1 function? In addition, TcPINK1 is constitutively active whereas the human enzyme may require activation. While certain aspects of PINK1 regulation (e.g. the substrate recognition) may be shared between TcPINK1 and hPINK1 there seems to be clear differences in the regulation of this enzyme in higher eukaryotes. Despite these limitations I think however that the presented structural data are very useful and informative and based on the interested in the role of PINK1 in PD the article may trigger functional studies on the human enzyme and it will be widely read. My recommendation is therefore to simply make the paper more transparent and to highlight the limitation of the current structural model more clearly in the revised version.

We have addressed these points separately above in the response to the Editor’s comments (comments 1-4)

Reviewer 3:[…] Major concern:Contrary to their claim in the Title and Abstract, the authors do not really reveal the activation mechanisms of hPINK1. They state that TcPINK1 is active in vitro (the crystal structure evidently shows why), but hPINK1 is not active in vitro. Why hPINK1 is not active and how it becomes activated are not explored.

We agree with the reviewer’s comments that the reasons for the inactivity of hPINK1 enzyme are still not clear and may be due to instability as also pointed by another referee. Therefore, we have removed the “activation mechanism” from the title and Abstract. We have also included new experiments in the revised manuscript including mutational analyses in hPINK1 enzyme as described above separately in response to Editor’s comments. We agree that understanding how hPINK1 is activated is interesting but further work will be required to decipher the activation mechanism for hPINK1 and is beyond the scope of this report.

Other substantive concerns:1) The authors speculate from the TcPINK1 crystal structure that Ins2 activates the kinase domain via a cyclin-CDK mechanism (involving αC). While this seems quite plausible based on the structure (and is arguably the most important feature of the structure), the authors did not perform any mutagenesis experiments to verify this.

We have now performed and included the assay results for testing the effect of Ins2 deletions on PINK1 activity in the revised manuscript. This is stated in more detail above in the response to the Editor’s comments (comments 1-4)

2) The authors speculate that Ins3, which is just upstream of the kinase domain and which is disordered in their crystal structure, interacts with ubiquitin to provide substrate specificity. To test this, they make a 10-residue deletion construct of Ins3 in TcPINK1 and show that ubiquitin phosphorylation is abrogated, whereas MBP phosphorylation (non-specific substrate, negative control) is less affected. As shown in Figure 3, though, MBP phosphorylation is down over 5-fold, indicating that the Ins3 deletion is probably affecting PINK1 stability, in addition to affecting (possibly) specific ubiquitin phosphorylation. A generic peptide substrate might serve as a better negative control.

We understand the reviewer’s concern about PINK1 stability, therefore, we have performed thermal denaturation assays on the mutants. The thermal shift assay data does not show any significant changes in the stability of mutants suggesting that the effect of Ins3 on substrate recognition is not due to instability of the mutant enzymes (Author response image 3). Furthermore, we have established that TcPINK1 can trans-phosphorylate other TcPINK1 molecules (kinase-inactive variants) and used this assay to show that Ins3 deletion does not impact on TcPINK1 trans-phosphorylation but does prevent ubiquitin phosphorylation. We think that the TcPINK1 trans-phosphorylation is more physiological than generic substrates and peptides and have presented this in the final analysis (Figure 2).

**Author response image 3. respfig3:** Thermal shift assay of TcPINK1 mutants